# Underwater Image Enhancement Based on Color Correction and Detail Enhancement

**Zeju Wu** [1,*], **Yang Ji** [1], **Lijun Song** [1] **and Jianyuan Sun** [2,3,*]

1  School of Information and Control Engineering, Qingdao University of Technology, Qingdao 266520, China
2  Department of Computer Science and Technology, Qingdao University, Qingdao 266071, China
3  Centre for Vision, Speech and Signal Processing, University of Surrey, Guildford GU2 7XH, UK
*  Correspondence: wuzeju@qut.edu.cn (Z.W.); jianyuan.sun@surrey.ac.uk (J.S.);
   Tel.: +86-178-0625-6941 (Z.W.); +86-178-6392-0436 (J.S.)

**Abstract:** To solve the problems of underwater image color deviation, low contrast, and blurred details, an algorithm based on color correction and detail enhancement is proposed. First, the improved nonlocal means denoising algorithm is used to denoise the underwater image. The combination of Gaussian weighted spatial distance and Gaussian weighted Euclidean distance is used as the index of nonlocal means denoising algorithm to measure the similarity of structural blocks. The improved algorithm can retain more edge features and texture information while maintaining noise reduction ability. Then, the improved U-Net is used for color correction. Introducing residual structure and attention mechanism into U-Net can effectively enhance feature extraction ability and prevent network degradation. Finally, a sharpening algorithm based on maximum a posteriori is proposed to enhance the image after color correction, which can increase the detailed information of the image without expanding the noise. The experimental results show that the proposed algorithm has a remarkable effect on underwater image enhancement.

**Keywords:** underwater image enhancement; U-Net; nonlocal means denoising; image sharpening

## 1. Introduction

Projects such as underwater biometrics [1], garbage collection [2], and resource exploration [3] all require clear images, but capturing a clear scene in the underwater world is not a simple task. There are a large number of media that can absorb light underwater; light will experience different degrees of attenuation when propagating underwater. Underwater images usually have problems such as color deviation, low contrast, and blurred details, which affect the subsequent research work on images. Therefore, it is necessary to study underwater image enhancement technology.

The current popular underwater image enhancement algorithms are divided into physical model algorithms and non-physical model algorithms. The algorithm based on the physical model mathematically models the degradation process of the underwater image and obtains a clear image by estimating the model parameters. Common physical model algorithms include the following. In 2010, Chao et al. [4] used the dark channel prior (DCP) to derive the background light and transmittance of the image and then restored the underwater image based on a physical model. In 2012, Chiang et al. [5] proposed a restoration method by compensating for the light absorbed by the underwater medium. Because there are a large number of media that can absorb light underwater according to the loss process of light underwater, a clear image is inverted by compensating light to achieve the effect of the restoration. In 2015, Galdran et al. [6] proposed an automatic red channel underwater image restoration method, which is an algorithm that combines the image defogging model with the attenuation rate of light in water.

The non-physical model algorithm does not consider the imaging process and model and improves the quality of the underwater image by image processing. Common al-

gorithms based on non-physical models are: In 2007, Iqbal et al. [7] proposed a sliding stretch method based on color space. The color correction is performed by histogram equalization [8] of the underwater image in RGB color space, and the contrast is adjusted by a stretching operation in the HIS color space. In 2012, Ancuti et al. [9] combined algorithms such as contrast enhancement, white balance, and extraction of regions of interest. These algorithms are used to enhance the underwater images, respectively, and then the processed images are fused according to a certain proportion. In 2013, Drews et al. [10] proposed the dark channel prior (UDCP) algorithm. The algorithm restores the underwater image by adjusting the blue and green channels of the image. In 2018, Peng et al. [11] used depth-dependent color change, scene light differentiation, and adaptive color correction to restore underwater images.

In recent years, deep learning [12] has been widely used in various fields and achieved great success. In 2018, Li et al. [13] applied the generative adversarial network [14] (GAN) to the problem of underwater image enhancement and generated a large number of datasets of underwater images and atmospheric images. In 2019, Li et al. [15] designed Water-Net, a neural network that enhances underwater images. In 2019, Wang et al. [16] proposed the UWGAN algorithm. GAN is used to generate a large number of datasets of underwater images and atmospheric images, and the medical image segmentation network U-Net [17] is used to train the model of mapping from underwater images to atmospheric images.

Many algorithms nowadays often lose the detailed information of the image in the process of color correction, and it is difficult to solve the problems of color deviation, low contrast, and detail loss of underwater images at the same time. To solve this problem, an algorithm based on color correction and detail enhancement is proposed. The algorithm transforms the underwater image enhancement problem into three problems: underwater image denoising, color correction, and detail enhancement, and designs algorithms for each problem. Experimental results show that the algorithm has a good effect on color correction and detail enhancement.

The innovations of this study are as follows:

(1) According to the underwater image noise problem, some improvements have been made to the nonlocal means denoising [17] (NL-means). The Gaussian weighted spatial distance and the Gaussian weighted Euclidean distance are combined as the index of NL-means to measure the similarity of structural blocks. The improved NL-means can preserve the texture features and edge information of underwater targets while maintaining the denoising function of the original algorithm.

(2) We use U-Net [18] to correct the color deviation of underwater images. Some improvements have been made to U-Net for underwater image problems. Introducing residual structure [19] and attention mechanism [20] into U-Net can effectively enhance feature extraction ability and prevent network degradation.

(3) According to the underwater image degradation model, an underwater image sharpening algorithm based on maximum a posteriori (MAP) is proposed. The algorithm can increase the detailed information of the image without expanding the noise.

## 2. Materials and Methods

This section studies the basic principle of light propagation in water and image enhancement from three aspects: image denoising, color correction, and image sharpening.

### 2.1. Underwater Image Imaging Model

The imaging of underwater images in the camera can be divided into direct component, backscattering component, and forward-scattering component [21].

(1) The direct component is the part where the light reaches the camera after being reflected by the target object in the process of underwater propagation. Its expression is:

$$E_d^c(x,y) = E^c(x,y)\exp[-a(c)d(x,y)] \tag{1}$$

where $(x, y)$ are the coordinates of image pixels, $c$ is the image of red, green, and blue three color channels, $E^c(x, y)$ is the reflected light of the target, $a(c)$ is the attenuation coefficient, and $d(x, y)$ is the distance between camera and target.

(2) The backscattering component is the part of light reaching the camera after scattering by the medium in water. Its expression is:

$$E_f^c(x, y) = g^c(x, y) \otimes E_d^c(x, y) \tag{2}$$

where $g^c(x, y)$ is the point-spread function, and $\otimes$ is the convolution operations.

(3) The forward-scattering component is the part where the light encounters the target object, is reflected by the target object, and then is scattered by the medium to reach the camera. The expression is:

$$E_b^c(x, y) = B_\infty(c)\{1 - \exp[-a(c)d(x, y)]\} \tag{3}$$

where $B_\infty(c)$ is the backlight.

### 2.2. Underwater Image Denoising

Due to the particularity of underwater impurities and illumination conditions, underwater images usually contain noise. NL-means can remove the noise of underwater images, but also reduce the texture features and edge information of images. The proposed algorithm makes some improvements to NL-means and combines Gaussian weighted spatial distance with Gaussian weighted Euclidean distance as a new index to measure the similarity of structural blocks. The improved NL-means can retain the texture features and edge information of underwater targets while maintaining the denoising function of the original algorithm.

### 2.2.1. NL-Means

NL-means is a good denoising algorithm. First, the entire image is divided into several regions. Then, it finds a region similar to it for each region. Finally, the average of these similar regions is used to replace the pixel values of these regions. The expression of NL-means is:

$$x_{i,j} = \frac{\sum_{k,g \in \Omega_{i,j}} \omega_{k,g,i,j} y_{k,g}}{\sum_{k,g \in \Omega_{i,j}} \omega_{k,g,i,j}} \tag{4}$$

where $\Omega_{i,j}$ is the local region centered on $(i, j)$, and the weight value $\omega_{k,g,i,j}$ is the distance between the image structure blocks at the exponential function mapping $(k, g)$ and $(i, j)$, which is expressed as:

$$\omega_{k,g,i,j} = \exp\left(\frac{-d\left(y_{k,g}, y_{i,j}\right)}{h^2 r^2}\right) \tag{5}$$

where $y_{i,j}$ is a vector centered on $(i, j)$, $h^2$ is the filter intensity coefficient, and $d\left(y_{k,g}, y_{i,j}\right)$ is the similarity measure of structural blocks, which is represented by Euclidean distance:

$$d\left(y_{k,g}, y_{i,j}\right) = \left\|y_{k,g} - y_{i,j}\right\|_2^2 \tag{6}$$

### 2.2.2. Improved NL-Means

The weighted kernel function of the NL-means algorithm can make the regions with high similarity obtain larger weights and the regions with low similarity obtain smaller weights. The ideal kernel function can output larger weights when the similar neighborhood distance is small, and the output decreases rapidly as the distance increases. Since the Gaussian kernel function increases less weight when the Euclidean distance is small, the image noise reduction performance will be reduced due to the frequent change of signal strength when it is used alone, which will affect the subsequent U-Net to extract the feature

information in the image. Inspired by the related noise reduction algorithm (Qiuyu Song, 2022) [22], the Gaussian weighted spatial distance and the Gaussian weighted Euclidean distance are combined as a new index to measure the similarity of structural blocks in the image, which effectively solves the problem of decreasing noise reduction ability.

The Gaussian weighted spatial distance can be expressed as:

$$d_s\left(y_{k,g}, y_{i,j}\right) = \left\|y_{k,g} - y_{i,j}\right\|^2 \tag{7}$$

The Gaussian weighted Euclidean distance can be expressed as:

$$d\left(y_{k,g}, y_{i,j}\right) = \left\|y_{k,g} - y_{i,j}\right\|_2^2 \tag{8}$$

Gaussian weighted spatial distance is combined with Gaussian weighted Euclidean distance as a new index to measure the similarity of structural blocks:

$$\omega_{k,g,i,j} = \exp\left(\frac{-\left\|y_{k,g} - y_{i,j}\right\|_2^2 \times \left\|y_{k,g} - y_{i,j}\right\|_2^2}{h^2 r^2}\right) \tag{9}$$

### 2.3. Underwater Image Color Correction

To solve the problem of color deviation of underwater images, an improved U-Net is proposed to fully extract the features of underwater images and adaptively learn the mapping relationship between underwater images and atmospheric images. The improved U-Net can increase the feature extraction ability and improve the network accuracy.

#### 2.3.1. U-Net

U-Net is an end-to-end network proposed by OlafRonneberger et al. [17] in 2015, which is mainly used for the semantic segmentation of medical images. U-Net is based on FCN [23] and only contains the convolution layer and pooling layer. The network has a fully symmetrical U-shaped structure with the same number of encoders and decoders on both sides. The model undergoes four down-samplings, and each coding layer contains two convolution operations to extract information from the image. Then down-sampling is performed by maximum pooling. The model also undergoes four up-sampling processes. First, concat the feature maps of the same depth using a jump connection. Then, the features in the image are extracted after two convolution operations. Finally, the features are passed up through up-sampling. Through the U-shaped structure, the down-sampling process extracts features, and the up-sampling process transmits features upwards so that the network extracts features with higher accuracy and better effect.

#### 2.3.2. Improved U-Net

As shown in Figure 1, the proposed underwater image color correction algorithm is based on U-Net and makes the following improvements to the network:

(1) Generally, deeper features can be extracted as the network deepens. However, due to the problem of network degradation, the effect of a deep network on feature extraction may not be as good as that of the shallow network. Changing the two convolution structures of each layer in the U-Net network to a residual structure can effectively prevent network degradation during color correction and ensure the ability of the network to extract features.

(2) Since U-Net has invalid semantic information when jumping connections, CBAM is added to the feature map of each layer in the encoder after passing through the convolutional layer. The feature map recovers the features of the image during up-sampling by CBAM. CBAM has a good resource allocation function, which can allocate more resources to more important features under the condition of limited resources, suppress other invalid features, and improve network accuracy.

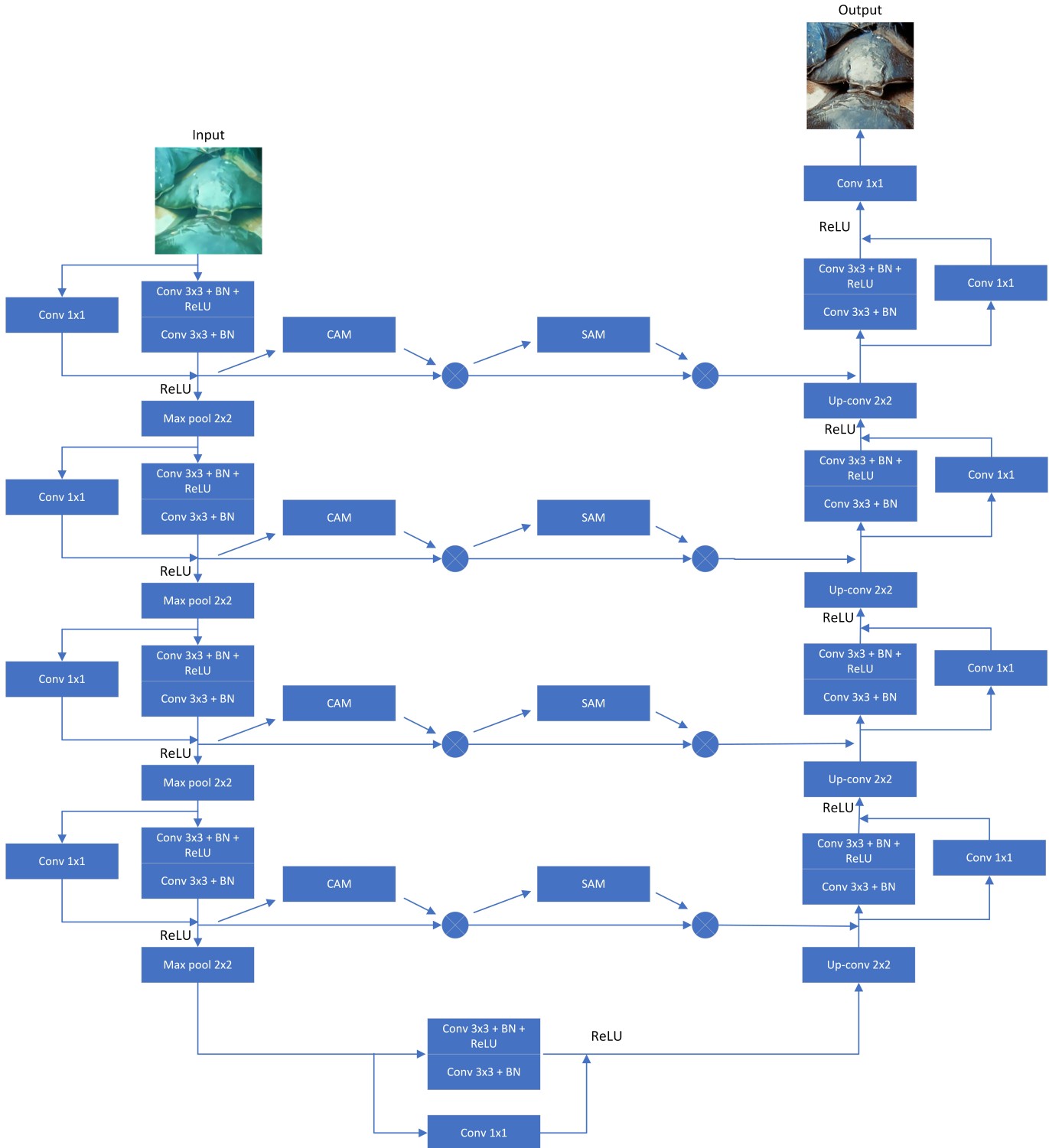

**Figure 1.** The structure of improved U-Net.

### 2.3.3. CBAM

CBAM is a very efficient attention mechanism including a channel attention module (CAM) and spatial attention module (SAM). As shown in Figure 2, the feature map goes through CAM and then SAM. CAM improves attention to the target category, and SAM improves attention to the target location.

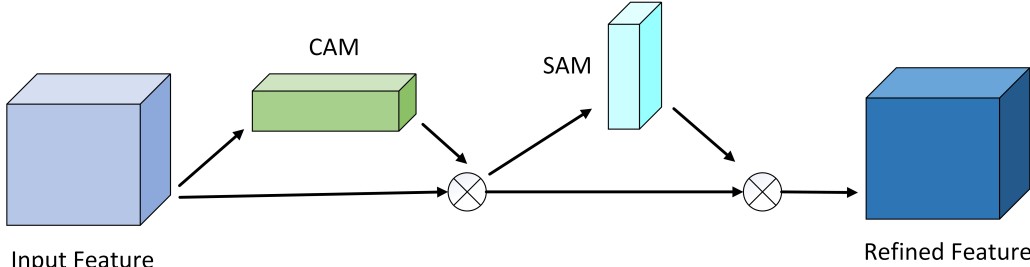

**Figure 2.** The structure of CBAM.

As shown in Figure 3, in CAM, the input features are first processed by maximum pooling and average pooling. In the backpropagation process, the maximum pooling is used to obtain the maximum value of each neighborhood, ignoring the influence of the non-maximum value. Average pooling calculates the average value of each neighborhood so that each pixel is involved in the calculation. Then the results are input into the shared fully connected layer to compress the spatial dimension of the features, and the obtained results are added at the pixel level. Then, the channel attention weight is obtained by the Sigmoid activation function. Finally, the channel attention weight is a matrix multiplied by the original image to improve the attention to the target category.

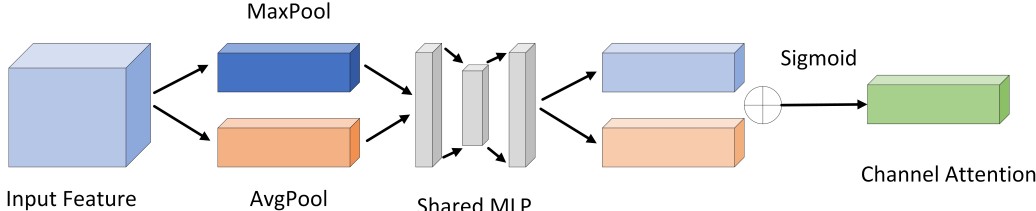

**Figure 3.** The structure of CAM.

As shown in Figure 4, in SAM, the input features are first processed by maximum pooling and average pooling. In the backpropagation process, the maximum pooling is used to obtain the maximum value of each neighborhood, ignoring the influence of the non-maximum value. Average pooling calculates the average value of each neighborhood so that each pixel is involved in the calculation. Then, concat the results of these two outputs at the channel level. Then, the result is subjected to a convolution operation to reduce the number of channels of the feature map. Next, the spatial attention weight is obtained by the Sigmoid activation function. Finally, the channel attention weight is a matrix multiplied by the original image to improve the attention to the target position.

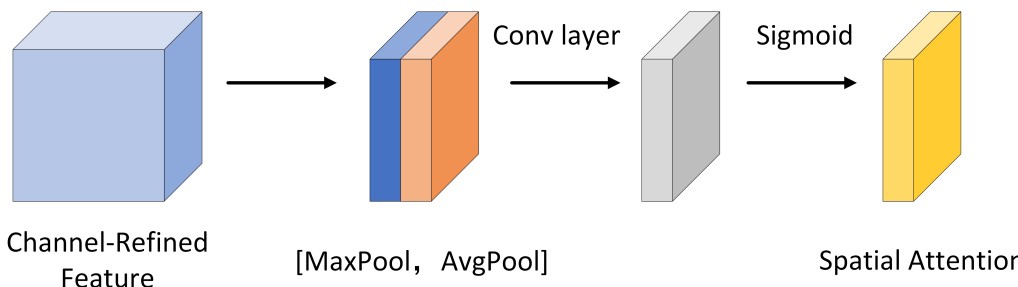

**Figure 4.** The structure of SAM.

## 2.4. Underwater Image Detail Enhancement

To solve the problem of lack of details in underwater images, this paper transforms the image detail enhancement process into the maximum likelihood problem of underwater

image deblurring probability model. The blurred kernel and the clear image are alternately updated by the MAP method to make the clear image closer to the real image.

### 2.4.1. Image Degradation Model

The image degradation model [24] is a model of the blurring process of the image underwater. The formula is as follows:

$$g(x,y) = k(x,y) \otimes f(x,y) + n(x,y) \tag{10}$$

where $f(x,y)$ is the input image, $k(x,y)$ is a degradation function, $n(x,y)$ is the random noise, and $g(x,y)$ is the blurred image.

If the noise interference is ignored, the model can be simplified as:

$$g(x,y) = k(x,y) \otimes f(x,y) \tag{11}$$

The usual solution to the image deblurring problem is: First, build a probabilistic model of image deblurring. Then, according to this model, the conditional probability density of the blurred image concerning the clear image and the degradation function is obtained. Finally, by maximizing this conditional probability, the most suitable clear image and degradation function are found. However, due to the particularity of underwater impurities and illumination conditions, underwater images usually contain a large amount of noise. While maximizing this conditional probability, the noise will also be amplified. This paper uses the MAP method to solve this problem. According to MAP, the image blur model can be expressed as:

$$\min_{f,k} F(g;k,f) + \alpha \rho_f(f) + \beta \rho_k(k) \tag{12}$$

where $F(\bullet)$ is the negative logarithm of the conditional probability density, representing the data item, $\rho_f$ and $\rho_k$ are priors for clear images and blur kernels, representing the regular term, and $\alpha$ and $\beta$ are the weight values of the regular term.

### 2.4.2. The Process of Sharpening Algorithm

As shown in Figure 5, the proposed underwater image sharpening algorithm flow mainly includes the blur kernel estimation part and the clear image estimation part. The update of the blur kernel and the clear image is an alternating process.

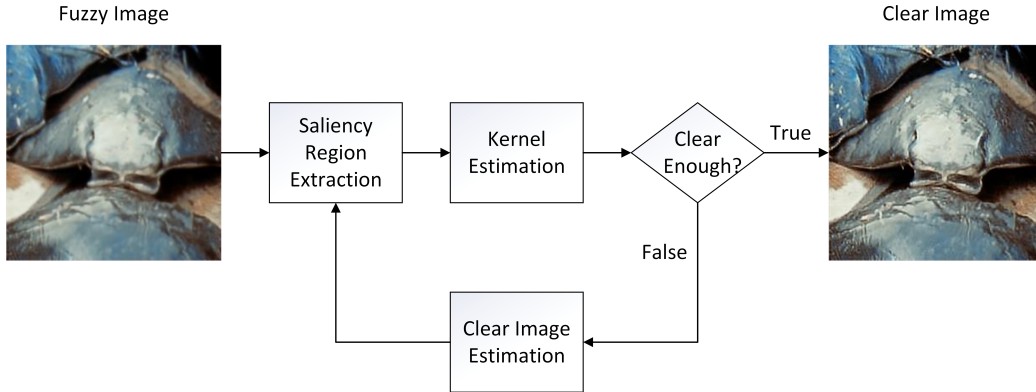

**Figure 5.** Flowchart of sharpening algorithm.

When updating the blur kernel, the current clear image is first blurred. Then, the adaptive threshold method is used to increase the salient region information of the image. Finally, the blur kernel is estimated according to our previously designed deblurring model.

When updating the clear image, the regularization term optimization model is first added to the deblurring model to increase the generalization ability of the model. Then, the

bilinear interpolation method is used to up-sample the clear image we estimate. Finally, the obtained clear image is used as the input to update the blur kernel, and the clear image and the blur kernel are continuously updated to make the clear image closer to the real image.

### 2.4.3. Saliency Region Extraction

The salient region refers to the contour information in the image. According to [25], effective extraction of salient regions of clear images can improve the accuracy of estimating blur kernels. It is a difficult task to extract the salient region of the image. This paper extracts the salient region from four aspects: brightness, color, direction, and edge [26].

Assuming that the three color channels of the input image are $r$, $g$, and $b$, the brightness feature can be expressed as:

$$I = (r + g + b)/3 \tag{13}$$

For color features, the saliency method of frequency domain harmonic [27] is used to obtain the color feature map of the image. First, we change the color to a uniform CIELab color space. Then, the transformed image is processed by Gaussian filtering. Finally, the square of the difference between the original image and the filtered image is used as the saliency map of the color. The formula is as follows:

$$C(x, y) = (I_u - I_{whc})^2 \tag{14}$$

where $I_u$ is the transformed image, and $I_{whc}$ is the filtered image.

For the directional features, Gabor filters [28] in four directions of 0°, 45°, 90°, and 135°are used to filter the grayscale images, respectively, and the feature maps in four directions are obtained. The expression is:

$$\text{sample}(x_0, y_0; \theta, \varphi) = \iint I(x, y) \, \text{Gabor}(x - x_0, y - y_0; \theta, \varphi) dx dy \tag{15}$$

where $X_0$ and $y_0$ are the center coordinates of the receptive field, and $I(x, y)$ is the input image.

For the edge features of the image, Canny edge detection [29] is used. First, Gaussian filtering is used to smooth the image and eliminate the noise in the image. Then, the gradient of the image is calculated to find possible edges. Then, non-maximum suppression is used to eliminate the edge of false detection. Finally, the double threshold method is used to filter out the edge information in the image.

### 2.4.4. The Estimation of Fuzzy Kernel

In the estimation process of fuzzy kernel to prevent nonconvex optimization problems and improve the estimation speed of the fuzzy kernel, this paper uses regularization terms to optimize the deblurring model. The expression is as follows:

$$\min_k \|\nabla S \otimes k - \nabla g\|_2^2 + \rho \|k\|_2^2 \tag{16}$$

where $\rho$ is used to balance the relative strength between data items and regular items.

By using the least square method, we can obtain:

$$k = F^{-1} \left( \frac{\bar{F}(\partial_x S) F(\partial_x g) + \bar{F}(\partial_y S) F(\partial_y g)}{F(\partial_x S)^2 + F(\partial_y S)^2 + \rho} \right) \tag{17}$$

where $F(\bullet)$ is the fast Fourier, $F^{-1}(\bullet)$ is the inverse transform, and $\bar{F}(\bullet)$ is the conjugate operation.

2.4.5. The Estimation of Clear Images

When estimating the clear image, we use the normalized prior term as the regularization term to optimize the equation. The formula is as follows:

$$\min_{f} \frac{\mu}{2} \|f \otimes k - g\|_2^2 + \frac{\|\nabla f\|_1}{\|\nabla f\|_2} \tag{18}$$

where $\nabla f$ is the gradient value of the clear image, and $\mu$ is the weight value of the data item.

## 3. Results and Discussion

The experiment was completed in the Python3.6 environment. The CPU is Intel (R) Core (TM) i5-6300 HQ CPU 8.00 GB RAM, using GPU acceleration, GPU is GTX1080, and the deep learning framework is PyTorch. A total of 2000 pairs of underwater image training networks were selected from the EUVP dataset. It covers images under different underwater scenes and lighting conditions. Adam [30] was used as the optimization algorithm during the training process, the batch size was set to 4, the initial learning rate was set to 0.001, and the learning rate was divided by 10 for every 30 epochs, eventually training 200 epochs.

To verify the effectiveness of the proposed algorithm, some representative images in the EUVP dataset were selected to compare the proposed algorithm with some classical algorithms in terms of color correction, detail enhancement, and image quality. These algorithms included IBLA [31], UDCP [10], ULAP [32], RGHS [33], Sea-thru [34], UWGAN [18], and FunieGAN [35].

### 3.1. Color Correction Experiment

To verify the effectiveness of the proposed algorithm for color correction, some images of blue and green scenes on the EUVP dataset were selected for color correction using the proposed algorithm and the classical algorithm.

As shown in Figure 6, (a), (b), (e), (f), (g) and (i) show the comparison results of images in the blues scene, and (c), (d), (h) and (j) show the comparison results of images in the green scene. It can be seen from the comparison results that the IBLA algorithm had a better enhancement effect for the image in the blue scene, and had a certain defogging effect for the image in the green scene, which can increase the contrast of the image, but at the same time cause the loss of image details. The UDCP algorithm can solve the fog blur problem of underwater images to a certain extent, but the processed images were darker overall, the contrast was low, and the effect was not ideal. The ULAP algorithm had a good enhancement effect on the image in the blue scene. The image processing in the green scene was reddish, and there was a certain color deviation from the real image. The RGHS algorithm had a certain solution to the problem of underwater image fog blur, but the overall effect was not obvious. The Sea-thru algorithm had a good effect on image enhancement in green scenes. For images in blue scenes, the details of the image can be increased, but the overall image color was blue, and there was a certain color deviation from the real image. The UWGAN algorithm had a good defogging effect for low contrast images while improving image contrast and increasing image details, but for high contrast images, the effect was not good. The FunieGAN algorithm had a certain effect of defogging and increasing image details, but the effect was not obvious. The algorithm of this study can effectively solve the fog blur problem of underwater images for images in blue and green scenes, and the image color was closer to the true value image. The results show that the proposed algorithm has a good ability to correct image color deviation for most underwater scene images.

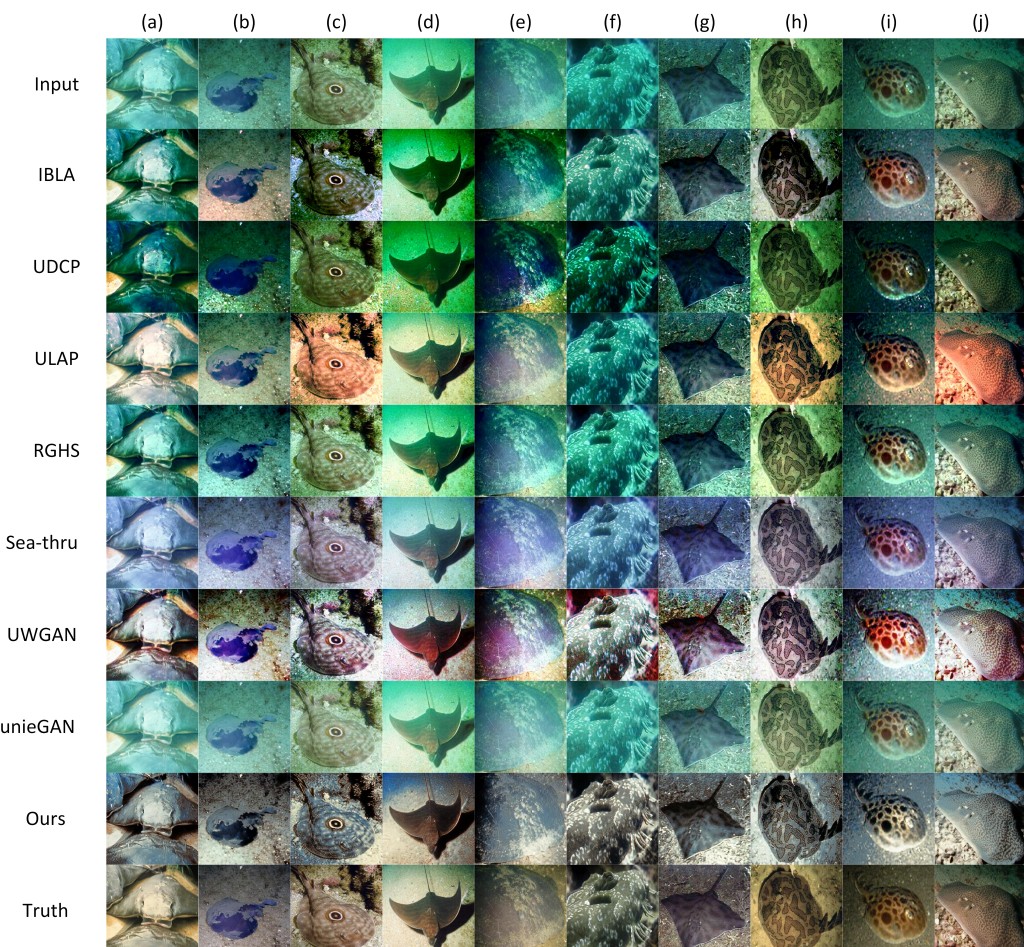

**Figure 6.** The comparison results of color correction effects between this research algorithm and other algorithms: (**a**,**b**,**e**–**g**,**i**) are blue scene images, (**c**,**d**,**h**,**j**) are green scene images.

### 3.2. Detail Enhancement Experiment

To verify that the proposed algorithm has the effect of enhancing image details, three images in the EUVP dataset were selected to compare the number of visible edges between the original image and the enhanced image. As shown in Figure 7, (a) is the edge detection of the original image and (b) is the edge detection of the enhanced image. The edges of the image contain a lot of information, and the number of edges in (b) is much higher than that in (a), which means that the proposed algorithm enhances more image details. Therefore, the proposed algorithm has the effect of increasing image details.

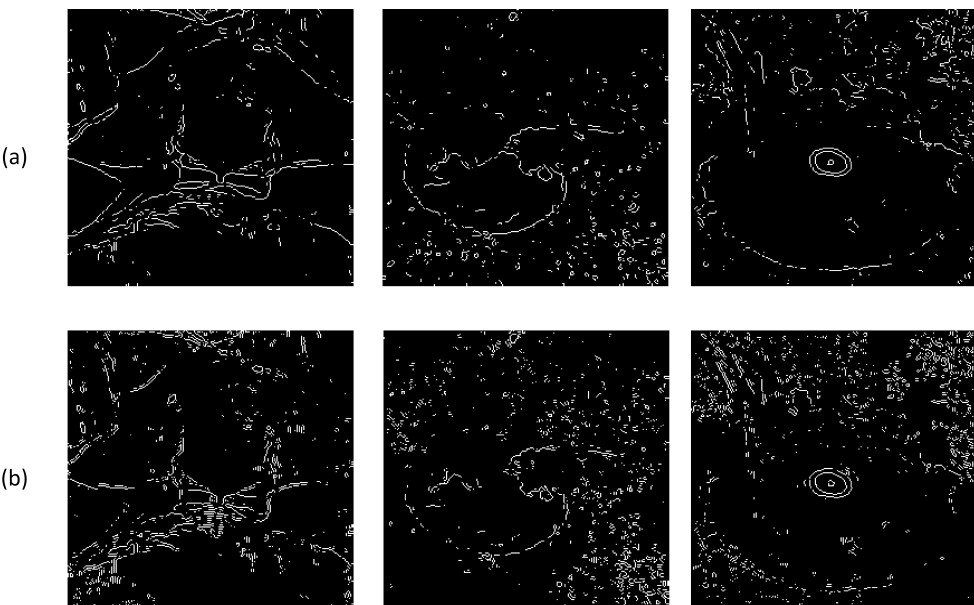

**Figure 7.** Visibility edge of gray: (**a**) original image; and (**b**) enhanced image by the proposed algorithm.

To verify the advantages of the proposed algorithm in detail enhancement, the visible edge growth rate of the image was used as the evaluation index to compare the proposed algorithm with the classical algorithm. Five underwater images were selected in the EUVP dataset, and the value of the visible edge growth rate e was calculated and compared. The visible edges contain a lot of detail information. The more the number of visible edge restoration means, the better the image enhancement effect. As shown in Table 1, the e values of IBLA and RGHS are lower, and the increase in the number of edges is not obvious. The e value of the proposed algorithm is the highest, which means that it is superior to other algorithms in detail enhancement. Therefore, the proposed algorithm can better restore the details of underwater images.

**Table 1.** Visible edge growth rate of different enhancement algorithms.

| Image | IBLA | UDCP | ULAP | RGHS | Sea-thru | UWGAN | FunieGAN | Ours |
|-------|------|------|------|------|----------|-------|----------|------|
| 1 | 1.452 | 1.477 | 1.343 | 1.242 | 1.502 | 1.607 | 1.622 | 1.375 |
| 2 | 1.405 | 1.331 | 1.272 | 0.992 | 1.422 | 1.507 | 1.423 | 1.221 |
| 3 | 0.792 | 2.220 | 1.721 | 1.523 | 2.204 | 2.332 | 2.215 | 2.274 |
| 4 | 0.541 | 0.605 | 1.023 | 1.005 | 1.652 | 1.552 | 1.476 | 1.775 |
| 5 | 1.305 | 1.427 | 1.275 | 1.121 | 1.445 | 1.307 | 1.502 | 2.513 |
| Average | 1.099 | 1.412 | 1.327 | 1.177 | 1.645 | 1.661 | 1.648 | 1.832 |

### 3.3. Image Quality Assessment

To verify the quality of the enhanced image, the peak signal-to-noise ratio (PSNR), structural similarity (SSIM), underwater image quality assessment metrics (UCIQE), and underwater image quality metrics (UIQM) [36] were used to quantitatively evaluate the measurement of the above ten images. PSNR is used to judge the degree of distortion between the enhanced image and the true value image. The greater the PSNR, the better the enhancement effect of the algorithm on the image. SSIM is an index of comprehensive contrast, brightness, and structural similarity. The larger the SSIM value, the more similar the enhanced image is to the true value image, which indicates that the effect of the algorithm is better. UCIQE is an index to judge the overall quality of the image. The larger the value, the better the effect of the algorithm. UIQM is an indicator of the color,

clarity, and contrast of the integrated image. The larger the value, the better the effect of the algorithm.

Table 2 shows the scores of SSIM, PSNR, UIQM, and UCIQE of various algorithms on the EUVP dataset. The UWGAN algorithm has the highest SSIM value and the best enhancement effect on the structural similarity of the image, but the UIQM value is very low, and the color and clarity are poor. The UCIQE value of the UDCP algorithm is the highest, and the overall image quality is high. However, the PSNR value is very low, and the image distortion is large. The PSNR and UIQM values of the proposed algorithm are higher than other algorithms, the distortion degree of the image is the smallest, and the enhanced image is closer to the original image. The algorithm proposed by SSIM and UCIQE values is not the highest, but it is superior to most algorithms. The results show that the overall quality of the image processed by the proposed algorithm is good, close to the real image, and in line with human visual senses.

**Table 2.** Underwater image quality evaluation of different enhancement algorithms.

| Scores | Input | IBLA | UDCP | ULAP | RGHS | Sea-thru | UWGAN | FunieGAN | Ours |
|---|---|---|---|---|---|---|---|---|---|
| SSIM | 0.794 | 0.694 | 0.579 | 0.756 | 0.759 | 0.804 | 0.827 | 0.779 | 0.745 |
| PSNR | 17.216 | 16.631 | 13.128 | 17.532 | 16.488 | 15.921 | 14.743 | 15.301 | 17.637 |
| UIQM | 1.377 | 2.772 | 2.987 | 2.270 | 2.208 | 1.066 | 1.875 | 2.240 | 4.035 |
| UCIQE | 0.379 | 0.459 | 0.497 | 0.452 | 0.447 | 0.378 | 0.476 | 0.431 | 0.429 |

*3.4. Running Time Experiment*

To verify the real-time performance of the algorithm, the running time of the proposed algorithm was compared with that of the classical algorithm. The experiment used the above ten images with a pixel size of $256 \times 256$. The running environment of the computer was Intel (R) Core (TM) i5-6300HQ CPU 8.00 GB RAM. All algorithms were tested on the same computer. IBLA, UDCP, ULAP, RGHS, and Sea-thru are traditional enhancement algorithms, and UWGAN and FunieGAN are deep learning algorithms. As shown in Table 3, the ULAP algorithm ran the fastest. The proposed algorithm is second only to ULAP and RGHS algorithms and is faster than other algorithms. The results show that the proposed algorithm has a faster running speed and can meet the real-time requirements of underwater image enhancement.

**Table 3.** Running time of different enhancement algorithms.

| Methods | IBLA | UDCP | ULAP | RGHS | Sea-thru | UWGAN | FunieGAN | Ours |
|---|---|---|---|---|---|---|---|---|
| Time (s) | 7.4774 | 3.1425 | 0.6091 | 1.4407 | 3.3012 | 1.5014 | 1.7256 | 1.4770 |

*3.5. Validation of Algorithm Effectiveness*

To verify the effectiveness of adding an attention mechanism and a residual module in the U-Net, U-Net and improved U-Net were trained using the EUVP dataset. Color correction was performed on the above 10 images using the trained results. SSIM, PSNR, UIQM, and UCIQE were used as evaluation indexes of correction effect.

As shown in Table 4, (a) is the correction effect of U-Net, (b) is the correction effect after adding the residual structure in U-Net, (c) is the correction effect of adding the attention mechanism in U-Net, and (d) is the correction effect of increasing both the attention mechanism and the residual structure. It can be seen that adding residual structure and attention mechanism in U-Net can effectively improve the SSIM, PSNR, UIQM, and UCIQE of images. Increasing the attention mechanism in U-Net works better than increasing the residual structure. Therefore, adding a residual structure and an attention mechanism in U-Net has a certain effect on the color correction of underwater images.

**Table 4.** The effect evaluation of improved U-Net.

| Network | SSIM | PSNR | UIQM | UCIQE |
|---|---|---|---|---|
| (a) | 0.703 | 17.521 | 1.422 | 0.383 |
| (b) | 0.711 | 17.755 | 1.451 | 0.385 |
| (c) | 0.723 | 17.968 | 1.570 | 0.390 |
| (d) | 0.731 | 18.201 | 1.782 | 0.392 |

*3.6. Ablation Study*

To understand the role of each component in the proposed algorithm, an ablation study was performed using the above ten images. SSIM, PSNR, UIQM, and UCIQE were used to measure the effect of each experiment. As shown in Table 5, (a) only the improved NL-means was used to denoise the underwater image, (b) only the improved U-Net was used to correct the color of the underwater image, (c) only the proposed sharpening algorithm was used to process the image, (d) the improved NL-means was used to denoise the underwater image and the improved U-Net was used to correct the color of the underwater image, and (e) the complete algorithm was used. The following conclusions can be drawn:

(1) Compared with the individual experimental results of each component, the complete model enhances the image best, which means that using these three components together is effective.

(2) The three components of the proposed algorithm have the effect of improving image performance. The improved U-Net has the greatest effect on improving SSIM and PSNR. The proposed sharpening algorithm has the greatest effect on improving UIQM and UCIQE.

**Table 5.** The quality evaluation of different components of the proposed algorithm.

| Experiment | SSIM | PSNR | UIQM | UCIQE |
|---|---|---|---|---|
| (a) | 0.724 | 17.415 | 1.427 | 0.388 |
| (b) | 0.731 | 18.201 | 1.782 | 0.392 |
| (c) | 0.716 | 17.338 | 1.845 | 0.401 |
| (d) | 0.733 | 18.272 | 2.329 | 0.413 |
| (e) | 0.745 | 18.637 | 4.035 | 0.429 |

*3.7. Application Test*

The proposed underwater image enhancement technology can be applied to underwater key point detection and image segmentation. To verify the effectiveness of the proposed algorithm, the underwater key point detection and image segmentation of the original image and the enhanced image were compared.

Underwater key point detection is an important technology for underwater image detection and recognition. To verify that the proposed algorithm has an enhanced effect on key point detection, SIFT key point detection [37] was performed on the original image and the enhanced image of the proposed algorithm, and the number of key points were compared. As shown in Figure 8, the number of key points of the enhanced image is much more than that of the original image. The results show that the proposed algorithm has a good effect on enhancing the details of underwater images, which is conducive to the detection and recognition of underwater images.

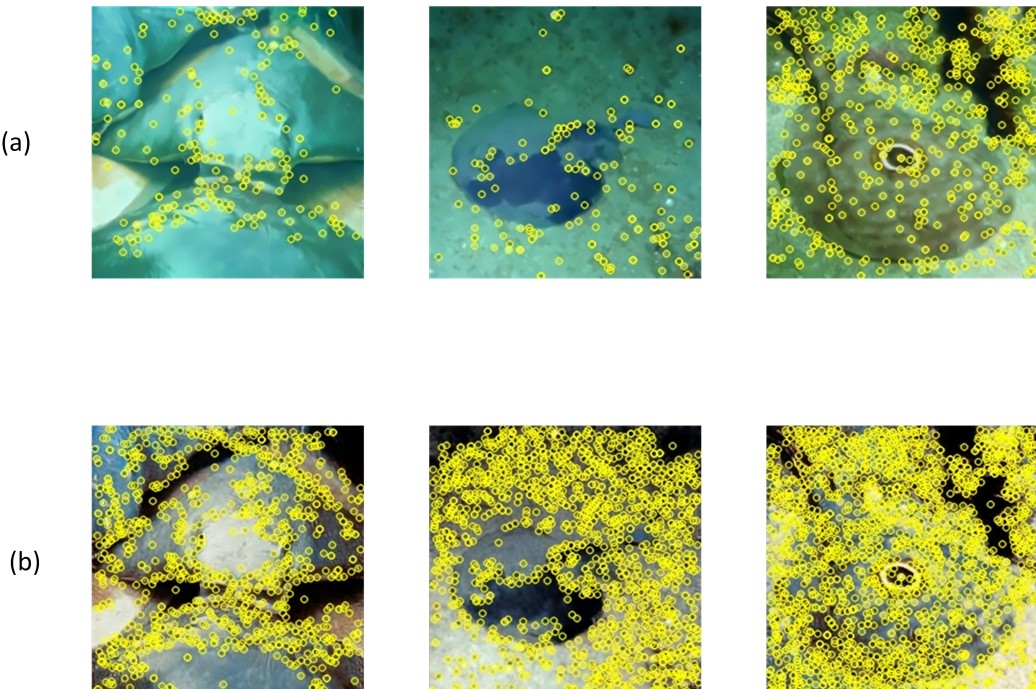

**Figure 8.** SIFT key point detection: (**a**) original image; (**b**) enhanced image by the proposed algorithm.

Underwater image segmentation divides the image into different regions according to the characteristics of the image. The fast FCM clustering algorithm [38] was used to segment the original image and the enhanced image. As shown in Figure 9, (a) is the segmentation effect of the original image, and (b) is the segmentation effect of the enhanced image. Using this algorithm to enhance the image, the segmentation effect is more accurate, especially for the segmentation of foreground and background. The results show that the proposed algorithm provides a more accurate segmentation effect for underwater image segmentation.

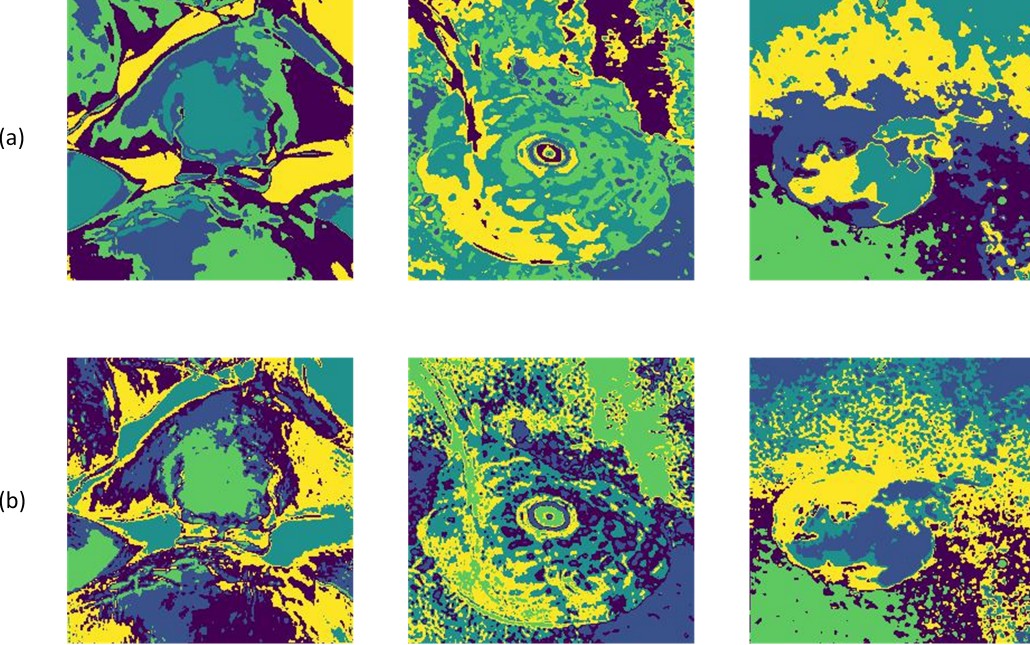

**Figure 9.** Underwater image segmentation: (**a**) original image; (**b**) enhanced image by the proposed algorithm.

## 4. Conclusions

In this work, a new enhancement algorithm was proposed. The algorithm enhances underwater images mainly from three aspects: image denoising, color correction, and detail enhancement. First, some improvements were made to the NL-means denoising algorithm. The improved NL-means can preserve edge and texture information while denoising. Then, the improved U-Net was used to correct the color of underwater images. Introducing residual structure and attention mechanism into U-Net can effectively enhance feature extraction ability and prevent network degradation. Finally, aiming at the problem of the lack of underwater image details, an image sharpening algorithm based on MAP was designed. The algorithm can increase image detail information without expanding noise.

The proposed algorithm was compared with some classical algorithms. The results show that the algorithm has a significant effect on the enhancement of underwater images, which provides help for future research of underwater image enhancement.

**Author Contributions:** Z.W. and Y.J. contributed to writing the original draft, revising and editing the manuscript, data collection, data analysis, statistics, and data interpretation; L.S. contributed to revising and editing the manuscript; J.S. contributed to conceptualization of the study, data interpretation, and revising and editing the manuscript. All authors approved the submitted version of the manuscript.

**Funding:** This work was supported by National Natural Science Foundation of China (No. 61501278).

**Institutional Review Board Statement:** Not Applicable.

**Informed Consent Statement:** Informed consent was obtained from all subjects involved in the study.

**Data Availability Statement:** The data presented in this study are available on request from the corresponding author.

**Conflicts of Interest:** The authors declare no conflict of interest.

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
