# Peer review of "Underwater Image Enhancement Based on Color Correction and Detail Enhancement"

_jmse, doi:10.3390/jmse10101513_

Round 1
Reviewer 1 Report
In this paper, an algorithm based on color correction and detail enhancement is proposed to solve the problems of underwater image color deviation, low contrast, and blurred details. From my view, this paper is well organized and the proposed method is valuable for this research filed. After reviewed this paper, there are some questions and suggestions as follows.
- You must review all significant similar works that have been done. Also, review some of the good recent works that have been done in this area and are more similar to your paper.
- It is necessary to experimentally analyze the proposed algorithm in terms of time consumed and compare with other algorithms.
- What are the advantages and disadvantages of this study compared to the existing studies in this area? This needs to be addressed explicitly and in a separate subsection.
- There are many grammatical mistakes and typo errors.
- Some final cosmetic comments:
* The results of your comparative study should be discussed in-depth and with more insightful comments on the behaviour of your algorithm on various case studies. Discussing results should not mean reading out the tables and figures once again.
* Avoid lumping references as in [x, y] and all other. Instead summarize the main contribution of each referenced paper in a separate sentence. For scientific and research papers, it is not necessary to give several references that say exactly the same. Anyway, that would be strange, since then what is innovative scientific contribution of referenced papers? For each thesis state only one reference.
* Avoid using first person.
* Avoid using abbreviations and acronyms in title, abstract, headings and highlights.
* Please avoid having heading after heading with nothing in between, either merge your headings or provide a small paragraph in between.
* The first time you use an acronym in the text, please write the full name and the acronym in parenthesis. Do not use acronyms in the title, abstract, chapter headings and highlights.
* The results should be further elaborated to show how they could be used for the real applications.
* Are all the images used in this work copyrights free? If not, have the authors obtained proper copyrights permission to re-use them? Please kindly clarify, and this is just to ensure all the figures are fine to be published in this work.
* Also, the list of references should be carefully checked to ensure consistency with between all references and their compliances with the journal policy on referencing.
Reviewer 2 Report
This paper proposes an underwater image enhancement approach. The proposed approach adds residual structure and attention mechanisms to U-Net. It also suggests a sharpening algorithm based on maximum a posteriori to enhance the restored image.
The introduction of residual structure and attention mechanisms into the conventional U-Net is straightforward. It lacks experimental results to verify this addition. Therefore, an ablation study is needed. It would be good to merge Figure 1 and Figure 6 together to provide both a clear U-Net structure and the proposed two modules.
1. Line 81: It states that "Because the Gaussian weighted Euclidean distance is too large, some edge and texture details are lost while reducing the image noise, which affects the subsequent U-Net network to extract the feature information in the image." Is there any evidence to justify this?
2. Line 67: Equation (4) can be reformated into a single line.
3. Please add reference numbers into Table 1. In addition, there is an "e" in Table 1. Is it a typo?
4. Figure 10 is duplicated from Table 2. If so, Figure 10 could be removed.
Reviewer 3 Report
The authors propose a method to restore underwater imagery considering the unique distortions specific to such environments. They compare their method against existing methods using standard metrics to establish the superiority of their method. This reviewer feels that the research shows promise, but the manuscript requires certain revisions to make it more suitable for this journal:
1. It seems some recent related works like https://doi.org/10.1007/s11042-020-10049-7 have been missed from the literature survey. Thus, revisiting the literature survey and making sure no important ideas have been missed would better help pinpoint the authors contributions.
2. It may be noted that real-time operation could be critical for many underwater image enhancement applications, and thus execution time in compute-constrained hardware would be a prime determining factor for suitability of the proposed method and those it is compared with. Thus, the authors are advised to report the execution time of the proposed method along with those it is compared with. Also, for a fair comparison, they should ensure that all methods are executed on the same hardware.
3. Although the overall standard of English usage in the manuscript is acceptable, it seems that there are certain instances where the authors could improve language usage: one such example is the use of the word "particularity" in point 2 in the list of contributions near the end of the Introduction section.
4. Where the authors first mention the use of U-Net in the Introduction section of when they go into more details in the proposed method, it may be beneficial to point out the more traditional uses of U-Net, specifically how it started as a way to segment medical images.
5. It seems from the qualitative results for the type of application being explored in the manuscript, subjective studies based on visualization using dynamic range conversion methods like https://doi.org/10.1007/978-3-030-04375-9_17 would generally improve the interest of other researchers in the reported results. However, in the interest of manuscript length, the authors could instead simply mention the potential of such studies in the conclusion or future works section of the revised manuscript.
6. Similarly, if manuscript length restrictions permit, the authors could consider adding a subsection under Results describing ablation studies. This is typical of papers that have multiple components in the proposed method, and readers and future researchers could be interested to know which of the several alterations to prior models contributed most to the performance improvements reported by the authors.
Round 2
Reviewer 1 Report
Good revisions have been made in the paper and the revised version has the necessary qualities for acceptance compared to the previous version. In my opinion, the article is acceptable in its current form.
Reviewer 2 Report
Thanks for the revision, that looks good to me.
Reviewer 3 Report
The authors seem to have addressed many of the comments raised in the previous round of reviews, and thus the manuscript may be accepted for publication.